# The E3 Ubiquitin Ligase Fbxo4 Functions as a Tumor Suppressor: Its Biological Importance and Therapeutic Perspectives

**DOI:** 10.3390/cancers14092133

**Published:** 2022-04-25

**Authors:** Shuo Qie

**Affiliations:** 1Department of Pathology, Tianjin Medical University Cancer Institute and Hospital, Huanhuxi Road, Tiyuanbei, Hexi District, Tianjin 300060, China; shuoqie@tmu.edu.cn; Tel.: +86-135-1292-5203; 2National Clinical Research Center for Cancer, Huanhuxi Road, Tiyuanbei, Hexi District, Tianjin 300060, China; 3Key Laboratory of Cancer Prevention and Therapy (Tianjin), Tianjin Medical University Cancer Institute and Hospital, Tianjin 300060, China; 4Tianjin’s Clinical Research Center for Cancer, Tianjin 300060, China

**Keywords:** Fbxo4, cyclin D1, Trf1, Fxr1, cell cycle, DNA damage response, metabolic reprogramming, cellular senescence, tumor development and progression

## Abstract

**Simple Summary:**

Fbxo4 is an E3 ubiquitin ligase that requires the formation of a complex with S-phase kinase-associated protein 1 and Cullin1 to catalyze the ubiquitylation of its substrates. Moreover, Fbxo4 depends on the existence of posttranslational modifications and/or co-factor to be activated to perform its biological functions. The well-known Fbxo4 substrates have oncogenic or oncogene-like activities, for example, cyclin D1, Trf1/Pin2, p53, Fxr1, Mcl-1, ICAM-1, and PPARγ; therefore, Fbxo4 is defined as a tumor suppressor. Biologically, Fbxo4 regulates cell cycle progression, DNA damage response, tumor metabolism, cellular senescence, metastasis and tumor cells’ response to chemotherapeutic compounds. Clinicopathologically, the expression of Fbxo4 is associated with patients’ prognosis depending on different tumor types. Regarding to its complicated regulation, more in-depth studies are encouraged to dissect the detailed molecular mechanisms to facilitate developing new treatment through targeting Fbxo4.

**Abstract:**

Fbxo4, also known as Fbx4, belongs to the F-box protein family with a conserved F-box domain. Fbxo4 can form a complex with S-phase kinase-associated protein 1 and Cullin1 to perform its biological functions. Several proteins are identified as Fbxo4 substrates, including cyclin D1, Trf1/Pin2, p53, Fxr1, Mcl-1, ICAM-1, and PPARγ. Those factors can regulate cell cycle progression, cell proliferation, survival/apoptosis, and migration/invasion, highlighting their oncogenic or oncogene-like activities. Therefore, Fbxo4 is defined as a tumor suppressor. The biological functions of Fbxo4 make it a potential candidate for developing new targeted therapies. This review summarizes the gene and protein structure of Fbxo4, the mechanisms of how its expression and activity are regulated, and its substrates, biological functions, and clinicopathological importance in human cancers.

## 1. Introduction

Proteins are the major molecules that participate in various biological processes and function as building blocks for the cells. The production and degradation of proteins are well-adjusted to maintain cellular homeostasis. In general, the cells regulate the synthesis of proteins through transcriptional and/or translational mechanisms [1]. Meanwhile, several different degradation pathways are employed to mediate protein degradation through proteasome, lysosome, or proteases [2,3,4,5]. Among these degradation mechanisms, ubiquitin-proteasome system (UPS) is the well-defined one, although there remains a plethora of unknown mechanistic details [6]. The UPS regulates the majority of cellular functions in eukaryotes, such as cell cycle progression, cell growth, differentiation, cell migration, and survival, through an irreversible reaction to degrade specific proteins in a selective manner [7]. In addition, Lysine 63-linked ubiquitylation can activate/inactivate downstream signaling pathways such as the Hippo signaling factor Yes-associated protein (YAP) or Protein kinase B (PKB/Akt) [8,9]. Therefore, the dysfunction of UPS not only affects protein stability, but also controls critical signaling pathways, leading to pathological conditions, e.g., inflammation, tumorigenesis and tumor progression [10].

The protein ubiquitylation is mainly conducted by three enzymes, including E1 (ubiquitin activating enzyme), E2 (ubiquitin conjugating enzyme), and E3 ubiquitin ligase [11,12,13]. The ubiquitin molecule is first activated by E1 using the energy from ATP hydrolysis, then transferred to E2, and finally ligated to the targets of E3 ubiquitin ligases [11]. According to their working mechanisms, E3 ubiquitin ligases are categorized into three groups: (1) Homologous to E6AP COOH-terminus (HECT)-type E3 enzymes requiring the formation of an E3-ubiquitin thioester intermediate prior to substrate ubiquitylation; (2) Really interesting new gene (RING) type E3s like finger-, U box- or plant homeodomain- E3s promoting the transfer of ubiquitin directly from E2s to their substrates; (3) RING-in-between-RING (RBR) E3s performing their function similar to that of the RING E3s [14]. The largest family of E3 ubiquitin ligases in mammalian cells is Cullin-RING ligase family, which is characterized by the formation of S-phase kinase-associated protein 1 (Skp1)−Cullin1 (Cul1)−F-box protein (SCF) ubiquitin ligase complexes. The SCF complexes employ RING-box protein 1 (Rbx1/Roc1) as a regulator of Cul1 [15]. F-box proteins (FBPs) receive their names because of the evolutionally conserved F-box domain. Generally, FBPs can be categorized into three sub-families: Fbxw with a WD40-repeat domain (12 members in total), Fbxl with a leucine-rich-repeat domain (21 members in total), and Fbxo containing F-box only with uncharacterized domains (36 members in total) [16].

Fbxo4, also known as Fbx4, belongs to the FBP family with a conserved F-box domain [17]. *F**BXO4* gene is located on chromosome 5p12 [18]. On the open-access online website (The Human Protein Atlas) [19,20], the single cell RNA sequencing analysis revealed that *F**BXO4* expression demonstrates low cell type specificity when comparing its expression in different cell types (Figure 1A,B), while the expression cluster of *F**BXO4* is mainly enriched for “Smooth muscle cells-Unknown function” (Figure 1C), suggesting that further studies are required to investigate the detailed functions of Fbxo4 in these cells. Several proteins are identified as Fbxo4 substrates, including cyclin D1, Telomeric repeat binding factor 1 (Trf1/Pin2), p53, Fragile X-related protein-1 (Fxr1), Myeloid cell leukaemia-1 (Mcl-1), Intercellular adhesion molecule-1 (ICAM-1), and Peroxisome proliferator-activated receptor gamma (PPARγ) [1,21,22,23,24,25,26,27,28,29]. This review summarizes the gene and protein structure of Fbxo4, the mechanisms of how its expression and activity are regulated, and its substrates, biological functions, and clinicopathological importance in human cancers.

## 2. The *F**BXO4* Gene and Protein Structure

FBPs have different isoforms that perform various biological functions [30,31,32]. Consistently, different Fbxo4 isoforms have also been elucidated by analyzing normal human liver/esophagus tissues and their relevant tumor cells (Figure 2A) [33]. In general, tumor cells have less wild type *F**BXO4* variant 1 compared to that in normal tissues. For example, gastric cancer cells mainly express variants 3 and 4 but not variant 1. The hepatocellular carcinoma tissues express more *F**BXO4* variants relative to that in normal counterparts. According to the transcriptional variants, Fbxo4 has four different protein isoforms including α, β, γ, and δ (Figure 2B). At the subcellular level, α isoform is mainly expressed in cytoplasm, while β, γ, and δ isoforms are generally located both in cytoplasm and nucleus [33]. Biologically, Fbxo4 isoforms perform different functions. For example, the ectopic expression of α isoform efficiently suppresses colony formation while β, γ, and δ isoforms not only promote colony formation in soft agar assay, but also enhance the migration of tumor cells.

As shown in Figure 2B, Fbxo4 molecule has four domains from N- to C-terminus: D domain for dimerization, F-box domain for the formation of SCF complex, Linker region and substrate binding domain [34]. Structural analyses combined with biophysical and biochemical techniques revealed the full-length Fbxo4 is deficient in E3 ubiquitin ligase activity when in monomeric state. The connecting loop between linker region and substrate binding domain is indispensable for dimerization. Additionally, the N-terminal domain can interact with Cul1-Rbx1 besides functioning for dimerization [35]. The screening of human esophageal squamous cell carcinoma (ESCC) tissues identified the existence of Fbxo4 mutations, such as S8R, S12L, P13S, L23Q, and G30N in D domain, and P76T mutation in F-box domain, leading to loss of function of Fbxo4 [21]. In addition, other mutations in the *FBXO4* gene have also been discovered in tumor tissues from the TCGA PanCancer Atlas Studies (Figure 3A), and in cells from the Cancer Cell Line Encyclopedia (Broad, 2019) on the cBioportal website (Figure 3B) [36,37]. In accordance, there are consistent mutations when comparing these bioinformatic findings from previous studies [21]. For example, the mutations located in the F-box domain can compromise the formation of SCF complex.

## 3. The Regulation of Fbxo4 Expression and Activity

As an E3 ubiquitin ligase, Fbxo4 is not simply regulated at the transcriptional level, but also by comprehensive and multi-dimensional mechanisms, e.g., expression by translational regulation, and activation by co-factor and post-translational modification.

### 3.1. Translational Regulation of Fbxo4

The regulation of Fbxo4 by Fxr1 is unexpectedly identified when genetically manipulating Fxr1 in mammalian cells [1]. Fbxo4 protein goes up without a corresponding increase of *F**BXO4* mRNA in cells upon Fxr1 knockdown, and vice versa. Further analyses identified Fxr1can interact with Fbxo4 mRNA via the AU-rich elements in its 3′ untranslated region [38,39], leading to reduced translation of Fbxo4. These findings suggest that the amplification of *FXR1* gene could be the initial hit for its overexpression, which is further enhanced by reducing the translation of its E3 ubiquitin ligase, Fbxo4.

### 3.2. αB-Crystallin Functions as a Co-Factor of Fbxo4

αB-crystallin, also termed as heat shock protein (Hsp) B5, belongs to small Hsp family that plays critical roles in protein homeostasis, for example, promoting protein folding and maintaining normal cellular functions [40]. Under stress conditions, αB-crystallin can bind to unfolded proteins, prevent their aggregation via its chaperone-like activities, and finally enhancing cellular ability to resist various stress conditions [41]. The yeast two-hybrid screening analysis identified αB-crystallin can interact with Fbxo4 depending on the phosphorylation of αB-crystallin at Serine (Ser)-19 and Ser-45. Moreover, the aggregation inducing αB-crystallin mutant, R102G, also facilitates its binding to Fbxo4 [42,43]. Biologically, the association of Fbxo4 with αB-crystallin enhances the ubiquitylation of detergent-insoluble proteins. Particularly, the phospho-mimic mutant αB-crystallin can recruit Fbxo4 to the SC35 speckles, highlighting their potentials to ubiquitylate the components of SC35 speckles [42]. Mass spectrometry analysis identified αB-crystallin is enriched in cyclin D1 immunoprecipitates [27]. Biochemical analysis revealed the formation of αB-crystallin, Fbxo4, and cyclin D1 complex. Fbxo4 can trigger the ubiquitylation of cyclin D1, and enhance its turnover; importantly, genetic manipulation of αB-crystallin is sufficient to alter cyclin D1 expression in mammalian cells [34,44,45]. p53 is another example that requires αB-crystallin for Fbox4-mediated ubiquitylation and degradation [24]. While this does not mean that αB-crystallin is always indispensable for Fbxo4 to perform its E3 ubiquitin ligase activities, for example, there is no direct evidence that αB-crystallin is required for Fbxo4-mediated ubiquitylation of Trf1, Fxr1, Mcl-1, ICAM-1, and PPARγ.

### 3.3. Phosphorylation and Dimerization of Fbox4

A plethora of studies revealed the majority of FBPs have to form dimers to perform their biological functions. Consistently, Fbxo4 is in the same condition when regarding to this point [21]. Biochemical analyses found the existence of N-terminal dimerization domain in Fbxo4 that regulates its E3 ubiquitin ligase activity. Mutational analyses revealed that Fbxo4 can be phosphorylated by Glycogen synthase kinase-3β (GSK-3β) at Ser-12 in human cells or Ser-11 in mouse cells [21]. The phosphorylation of those Ser residues provides a docking site for 14-3-3ε that facilitates the homodimerization and activation of Fbxo4 (Figure 2B and Figure 3A,B) [46]. 

## 4. The Identified Substrates of Fbxo4

The first step to identify a new substrate of an E3 ubiquitin ligase is to screen the interacting candidates through immunoprecipitation. As a biomedical interaction repository database, the BioGRID provides some clues for identifying potential substrates of Fbxo4 (Figure 4). By analyzing the BioGRID database [47], several proteins are found to be identified as Fbxo4 substrates including cyclin D1, Trf1/Pin2, p53, Fxr1, Mcl-1, ICAM-1 and PPARγ. Particularly, cyclin D1 is the most investigated one with well-defined mechanisms for ubiquitylation and degradation. The following is a detailed summarization of how those substrates are regulated by Fbxo4.

### 4.1. Cyclin D1

Cyclin D1 belongs to the D cyclin family that functions as a key regulator of cell cycle progression, specifically for the transition from G_1_ to S phase [48,49,50]. Biologically, cyclin D1 can form a kinase complex with cyclin-dependent kinase 4 or 6 (CDK4/6) that phosphorylates and inactivates of Retinoblastoma (Rb) protein, leading to the release of E2F transcriptional factors to initiate the expression of downstream genes to drive cell cycle progression [7,17]. Therefore, cyclin D1 is regarded as one master regulator of cell proliferation, and its expression is usually dysregulated in human cancers. The overexpression/activation of oncogenes or the loss/inactivation of tumor suppressors promotes tumor development and progression via the coordination of D cyclins like cyclin D1 to perform their biological functions. Hence, it is not surprising that oncogenes, tumor suppressors, or their relevant downstream signaling can regulate cyclin D1 expression [7,17]. In addition, a bunch of physiological stimuli also enhance cyclin D1 expression, e.g., growth factors, nutrient availability, and the presence of extracellular adhesion signaling [51]. Cyclin D1 is highly expressed in many human cancers including pancreatic adenocarcinoma, lung cancer, breast cancer, head and neck squamous cell carcinoma (HNSCC), cutaneous melanoma, endometrial cancer, colorectal carcinoma, ESCC, mantle cell lymphoma, and so forth [52,53,54,55,56,57,58,59,60,61,62,63,64,65,66]. Clinically, cyclin D1 expression is correlated with tumor size, invasion, metastasis and clinical stages, highlighting that cyclin D1 can be considered as a prognostic factor to evaluate patient prognosis, and cyclin D1 signaling can be targeted to treat human cancers [67].

The protein level of cyclin D1 oscillates in a cell cycle-dependent manner. In the presence of growth factors, cyclin D1 levels are induced to drive cell cycle progression through G_1_ phase; after passing through this checkpoint, cyclin D1 expression is rapidly descending in S phase, indicating the existence of efficient mechanisms to degrade cyclin D1. Threonine (Thr)-286 is critical for cyclin D1 degradation because the phosphorylation of Thr-286 facilitates the transportation of cyclin D1 from nucleus to cytoplasm where it is poly-ubiquitylated and degraded [68,69]. The Thr-286 mutations have been identified in human cancer specimens, and importantly, the tumorigenic isoform of cyclin D1, cyclin D1b, is also found in human cancers from esophagus, breast, lung, and prostate as well as in lymphoma. Cyclin D1b shows increased protein stability because this isoform does not have a C-terminus that facilitates its degradation [70,71].

The protein stability of Cyclin D1 is regulated by several E3 ubiquitin ligases, such as Fbxo4, Fbxo31, Fbxw8, β-transducin repeat-containing protein (β-TrCP), Anaphase-promoting complex/cyclosome (APC/C), and Activating molecule in Beclin 1 regulated autophagy (AMBRA1) [27,72,73,74,75,76,77,78]. Fbxo4 is the first identified E3 ubiquitin ligase that promotes cyclin D1 degradation. As the primary kinase, GSK-3β phosphorylates cyclin D1 at Thr-286, which facilitates its re-localization in the cytoplasm [27]. Meanwhile, GSK-3β also phosphorylates Fbxo4 at Ser-12/Ser-11, promotes its homodimerization, and activates the E3 ubiquitin ligase activity. Thereafter, SCF-Fbxo4 complex catalyzes the poly-ubiquitylation of cyclin D1 and targets it to proteasome for degradation (Figure 5) [21]. The genetic manipulation of Fbxo4 successfully alters the stability of cyclin D1 protein, which is evidenced by the accumulation of cyclin D1 protein in human ESCC tissues and melanoma cells with defective Fbxo4 [79]. For example, the presence of Fbxo4 mutations disrupts its ability to dimerize and to form the SCF complex in ESCC tumor specimens, and the existence of a substrate-binding deficient Fbxo4, Isoleucine (Ile)-377 to methionine mutant, in melanoma cells [79].

### 4.2. Trf1/Pin2

In eukaryotic cells, telomere defines the nucleoprotein complex that maintains the length of linear chromosomes. The telomeric DNA is composed by repetitive sequence including double-strand tandem repeats and single-strand G-rich overhang that are associated with the shelterin complex, particularly Trf1 functions as one key component [80]. Structurally, Trf1 has four domains, including an N-terminus, a dimerization domain (also known as TRF homology domain), a linker region, and a C-terminal DNA-binding domain [81]. Loss of Trf1 is embryonically lethal because of its importance in maintaining telomere replication and protecting telomere length. Ectopic expression of Trf1 induces the telomere shortening and vice versa, suggesting Trf1 compromises the telomerase-dependent telomere extension. Biologically, Trf1 plays critical roles in regulating cell division and cellular response to DNA damage [82].

The association of Trf1 to telomeric DNA is a dynamic process that is testified by the interchange of bound Trf1 to free Trf1. After the dissociation from telomeric DNA, Trf1 undergoes fast turnover depending on poly-ubiquitylation. Moreover, Trf1 protein levels are fluctuating across the cell cycle with an obvious accumulation during the G_2_-M transition [23,29]. Trf1 ubiquitylation is mediated by three E3 ubiquitin ligases, including Fbxo4, β-TrCP, Ring Finger Protein, and LIM Domain Interacting (RLIM) [23,26,83]. Fbxo4 is first reported to interact with Trf1 in both yeast and mammalian cells. Biochemical analysis revealed TRFH domain of Trf1 acts as the interacting domain with Fbxo4, interestingly, this domain is also the docking site for Trf1-interacting nuclear factor 2 (TIN2) that can compete with Fbxo4 and compromise Trf1 ubiquitylation [81]. As an essential pre-mRNA splicing factor, U2 small nuclear ribonucleoprotein (snRNP) auxiliary factor 65 (U2AF65) can interact with Trf1 in vitro and in vivo to stabilize Trf1 protein through interrupting Fbxo4-mediated ubiquitylation and degradation [84]. Biochemically, the ectopic expression of Fbxo4 reduces the half-life of Trf1 protein, while the loss of Fbox4 expression or function leads to the accumulation of Trf1 and telomere shortening. 

### 4.3. p53

The p53 protein, encoded by the *TP53* gene, functions as a transcription factor with tumor suppressor activities [85]. Under physiological conditions, p53 regulates a bunch of cellular processes, such as controlling cell division, maintaining genomic stability through DNA damage response, inducing apoptosis, regulating autophagy, and immune response [86]. Therefore, a plethora of p53 mutations are observed in human cancers derived from the breast, colon, lung, liver, prostate, bladder, and skin [87]. Under pathological conditions, mutated p53 loses its capacity to arrest cell cycle and to repair DNA damage, leading to the replication of damaged DNA, cell proliferation and tumorigenesis.

Heat shock factor 1 (Hsf1) is transcriptionally upregulated and plays important roles upon exposing to a variety of stresses like DNA damage, nutrient withdrawal, and oncogenic activation [88]. The activation of Hsf1 upregulates heat shock proteins (Hsps) like Hsp27 and αB-crystallin [24,89]. αB-crystallin facilitates the formation of a complex with p53 and Fbxo4. Therefore, the accumulation of p53 protein in cells with deficient Hsf1 is the result of reduced protein ubiquitylation and degradation-mediated by Fbxo4. 

### 4.4. Fxr1

Fxr1 is one member of the fragile X-related protein family that contains nuclear localization signal, nuclear export signal, Tudor domain, K homology domain and RGG box [90]. Fxr1 can interact with RNAs, direct their transportation, and regulate their metabolism. Under physiological conditions, Fxr1 is mainly expressed in brain tissues like oligodendrocytes, microglia, and endothelial cells in cortex [91], highlighting its role in regulating the functions of central nervous systems. Recent studies provide more clues on the roles of Fxr1 in human tumors, for example, elevated Fxr1 levels are detected in tumor tissues relative to that in normal tissues, and its expression is correlated with patients’ prognosis in lung cancer, breast cancer, head and neck cancer, and oral cancer [92,93]. Biological investigation revealed overexpression of Fxr1 leads to increased cell proliferation through compromising the expression of cell cycle inhibitors such as p21^Cip1^ and p27^Kip1^ [1,93]. The co-amplification of *F**XR1*, protein kinase C, iota (*PRKCI*), and epithelial cell transforming 2 (*ECT2*) is detected in lung squamous cell carcinoma (LSCC) tissues; biologically, Fxr1 forms a complex with PRKCI and ECT2 to promote cell proliferation and invasion [93]. In addition, Fxr1 stabilizes c-Myc mRNA through binding to the AU-rich elements in its 3′ untranslated region. Meanwhile, Fxr1 also interacts with eIF4A1 and eIF4E, and recruits the translation initiation complex to promote the translation of c-Myc mRNA [94].

Mass spectrometry analysis identified Fxr1 is a binding partner of Fbxo4 [1]. Further biochemical analysis confirms the interaction between these two proteins. Particularly, the C-terminus of Fbxo4 is indispensable for interacting with Fxr1. Mutational analyses demonstrated that Glutamate (Glu)-379, -380 and Ile-377 in Fbxo4 is indispensable for interacting with Fxr1; and Valine (Val) 178 in Fxr1 is required for binding to Fbxo4. In the presence of GSK-3β, Fbxo4 enhances the ubiquitylation of Fxr1 in vitro and in vivo, and finally directs it to proteasome for degradation. Consistently, altered Fxr1 protein stability was also observed when overexpressing Fbxo4 mutants harbored different binding activities to Fxr1.

### 4.5. Mcl-1

Mcl-1, belonging to the Bcl-2 family, functions as a pro-survival factor that can inhibit apoptosis through compromising pro-apoptotic members, such as Bim, Bax, and Bak [95]. Mcl-1 is a labile protein with a very short half-life. This characteristic is associated with its biological function in regulating cell survival/apoptosis under stress conditions. Mcl-1 is upregulated in a plethora of human tumors, such as cancers of the lung, breast, pancreases, prostate, cervix, ovary, lymphoma, and leukemia [96]. The overexpression of Mcl-1 also induces acquired resistance to several chemotherapeutic compounds, such as cisplatin, paclitaxel, gemcitabine, and vincristine [97,98,99,100]. Mcl-1 can be degraded by several E3 ubiquitin ligases, including Mcl-1 ubiquitin E3 ligase (MULE/ARF-BP1), Tripartite Motif-containing protein 17 (TRIM17), Parkin, β-TrCP, Fbxw7, and APC/C [101]. In non-small cell lung cancer (NSCLC) cells, Fbxo4 protein levels are reversely correlated with Mcl-1 expression [22]. As an E3 ubiquitin ligase, Fbxo4 can ubiquitylate and mediate the degradation of Mcl-1, resulting in increased sensitivity to apoptotic inducers. Particularly, overexpression of Fbxo4 effectively enhances the sensitivity of NSCLC cells to chemotherapeutic compounds, including cisplatin or paclitaxel, highlighting the therapeutic potential by targeting Fbxo4-Mcl-1 signaling for treating NSCLC.

### 4.6. ICAM-1

ICAM-1 is a glycoprotein located in cellular membrane and functions as an adhesion receptor that regulates inflammatory response via recruiting leukocytes from the blood stream [102]. ICAM-1 is expressed at a low basal level in endothelial, epithelial, and immune cells, while it can be upregulated by inflammatory cytokines to mediate traffic of leukocytes, and the barrier function of endothelial and epithelial cells [103]. In human tumors, ICAM-1 is expressed in all cell types in the tumor microenvironment. Biologically, ICAM-1 facilitates tumor development and progression. As a downstream factor of extracellular signal-related kinase, ICAM-1 plays a role in regulating the metastasis of human breast cancer cells. Further screening identified that Fbxo4, as a specific E3 ubiquitin ligase, can interact with ICAM-1 through binding to its intracellular region, finally leading to the ubiquitylation and degradation of ICAM-1 [25].

### 4.7. PPARγ

PPARγ is a member of the nuclear hormone receptor superfamily that performs their biological functions via binding to both natural and synthetic ligands [104]. PPARγ regulates cell proliferation, differentiation, survival, and apoptosis, mainly through affecting the metabolism of lipids and carbohydrates [105]. Physiologically, PPARγ not only regulates adipogenic and lipogenic pathways in white adipocytes, but also controls the thermogenic processes in brown or beige adipocytes [106]. Hsp20 is one of the most upregulated genes in human adipose-derived stem cells [107]. A recent study found that the loss of Hsp20 enhances the whole-body energy expenditure and increases the fat mass in transgenic mice. Further experiments revealed PPARγ contributes to Hsp20-mediated lipogenesis and thermogenesis. Mechanistically, Hsp20 forms a complex with Fbxo4 to promote the ubiquitylation and degradation of PPARγ [28].

## 5. The Clinicopathological Importance of Fbxo4

The well-established Fbxo4 substrates are found to regulate cell cycle progression, cell proliferation, survival/apoptosis, and migration/invasion, highlighting these factors possess oncogenic or oncogene-like activities (Table 1). As such, their E3 ubiquitin ligase, Fbxo4, is defined as a tumor suppressor [108]. Several studies found that Fbxo4 expression is downregulated in hepatocellular carcinoma (HCC) and HNSCC tissues comparing to their normal counterparts, and Fbxo4 loss of function mutations are revealed in human ESCC tissues and melanoma cells [1,21,33,79]. Moreover, low Fbxo4 levels are associated with poor prognosis in patients with breast cancer [25,109,110]. In accordance, Fbxo4 knockout mice can photocopy its tumor suppressor activities in Braf^V600E/+^ melanoma, and esophageal and forestomach papilloma models [79,111], further confirming that Fbxo4 has tumor suppressor activities.

However, the ENCORI analyses of the TCGA database indicate Fbxo4 could play more complicated roles in human tumors when only considering Fbxo4 mRNA levels [113]. For example, reduced Fbxo4 expression is correlated with poor prognosis in patients with skin cutaneous melanoma (SKCM), rectum adenocarcinoma (READ), cervical squamous cell carcinoma and endocervical adenocarcinoma (CESC), and sarcoma (SARC), but low Fbxo4 expression is associated with good prognosis in patients with brain lower grade glioma (LGG) and uveal melanoma (UVM) (Figure 6A–G). Several potential explanations for this discrepancy include: (1) Besides transcriptional regulation, Fbxo4 activity is another critical dimension that pertains to its biological functions. For example, Fbxo4 needs to be activated by co-factor, phosphorylation, and dimerization to perform its functions [21,27,42,43]. Hence, only considering Fbxo4 mRNA levels may overlook these critical mechanisms. (2) The TCGA database does not analyze the expression of various transcript variants or isoforms of Fbxo4, which possess totally opposite biological functions on cell proliferation and migration [33]. (3) The degradation of several substrates of Fbxo4 causes genome instability that leads to even high stage diseases and worse prognosis. For example, the degradation of Trf1 that maintains telomere replication and protects telomere length [26], or the degradation of p53 that directly mediates the DNA damage response never mentioning the elusive effect of Fbxo4 in the background of mutant p53 [24]. Therefore, further studies are required to identify new substrates or to dissect different mechanisms that control Fbxo4 activity in different human tumors. The patients will be benefited only after the whole picture is depicted concerning how Fbxo4 and its relevant signaling regulate the development and progression of human cancers, because the biological functions of Fbxo4 are mainly correlated with and determined by the activities of its substrates.

## 6. The Biological Functions of Fbxo4

### 6.1. Cell Cycle

As an E3 ubiquitin ligase of cyclin D1, Fbxo4 regulates cell cycle progression in both normal and tumor cells [21]. Knockdown of Fbxo4 drives an obvious accelerated entry from G_1_ into S phase because the accumulation of cyclin D1 in cells with defective Fbxo4 functions [114]. In accordance, loss of Fbxo4 efficiently promotes cell proliferation as well as cellular transformation upon the induction of RasV12 [109]. Not surprisingly, the observations from transgenic mice also confirm the tumor suppressor activities of Fbxo4. The transgenic mice with Fbxo4−/− or Fbxo4−/+ genotypes can spontaneously develop various pathological conditions, including lymphoblastic lymphoma, extramedullary hematopoiesis/intense myeloid proliferation, hemangioma/angioinvasive tumor, dendritic cell tumor, histiocytic sarcoma, early myeloid tumor, mammary carcinoma, and uterine tumor [109]. In addition, loss of Fbxo4 enhances the development of aggressive melanoma in a Braf^V600E/+^ mouse model [79]. Furthermore, Fbxo4 loss also facilitates the development of esophageal and forestomach papilloma-induced by N-nitrosomethylbenzylamine in transgenic mice that can be therapeutically treated by CDK4/6 inhibitor, palbociclib, suggesting that Fbxo4 can be considered as a marker to assess patients’ response [112].

### 6.2. DNA Damage Response

Fbxo4 controls DNA damage response because of inappropriate expression of cyclin D1 and Trf1 upon genomic damages. The ectopic expression of a constitutively nuclear mutant of cyclin D1 stabilizes the DNA replication licensing factor, Chromatin Licensing and DNA Replication Factor 1 (Cdt1) [115,116]. The dysregulation of Cdt1 promotes DNA re-replication and DNA damage, leading to chromatid breaks that favor the occurrence of second hit and latterly tumor development. Consistently, defective Fbxo4 activation also accumulates cyclin D1 in the nucleus and enhances the dysregulation of Cdt1 [115]. As a member of the Never in Mitosis Gene A kinase family, NIMA Related Kinase 7 (NEK7) functions as a regulator of telomere integrity. Upon DNA damage, Nek7 is recruited to telomeres to phosphorylate Trf1 at Ser-114 that compromises the interaction between Trf1 and Fbxo4, reducing the ubiquitylation and degradation of Trf1 to alleviate the DNA damage response and to maintain telomere integrity [82].

### 6.3. Tumor Metabolism

As a critical nutrient for highly proliferating tumor cells, glutamine can be utilized as an energetic, biosynthetic, or reductive precursor [117,118,119]. Glutamine-addiction defines the phenomenon wherein cells depend on glutamine for survival and proliferation. Generally, these cells demonstrate elevated glutaminase 1 (GLS1) expression and glutamine uptake [120]. GLS1 is upregulated in a variety of human tumors, and increased expression or elevated enzymatic activity of GLS1 is significantly associated with poor prognosis, indicating GLS1 is a point of interest for developing targeted therapies [121]. Previous studies revealed dysregulation of Fbxo4-cyclin D1 signaling results in defective mitochondrial function and reduced energy production. Particularly, loss of Fbxo4 promotes the dependency of normal cells and ESCC cells on glutamine. Therefore, combined treatment with CB-839/telaglenastat (GLS1 inhibitor) plus metformin can effectively suppress ESCC cell proliferation in vitro and xenograft growth in vivo [51]. Palbociclib-resistant ESCC cells demonstrate GLS1 upregulation is in a c-Myc-independent manner, making them hyper-glutamine-addicted and more sensitive to combined treatment than their parental counterparts. These findings provide a novel second-line therapy to overcome acquired resistance to CDK4/6 inhibitors.

### 6.4. Cellular Senescence

Cellular senescence defines the arrest of cell proliferation of diploid cells as a result of telomere shortening [122]. Morphologically, senescent cells become enlarged and flattened with the accumulation of lysosomes and mitochondria [123]. The replicative stress enhances DNA damage response that activates the transcription of *TP53* and the induction of p21^Cip1^. Additionally, telomere shortening promotes the expression of p16^Ink4a^ [124]. Both p21^Cip1^ and p16^Ink4a^ can cause cell cycle arrest at the G_1_-S phase. Normally, senescent cells are characterized by the senescence-associated secretory phenotype (SASP) that refers to the secretion of a group of growth factors, chemokines, and cytokines. As an RNA-binding protein, Fxr1 can bind to p21^Cip1^ mRNA through the G-quadruplex structure in its 3′ untranslated region and promote the degradation of p21^Cip1^ mRNA. Therefore, silencing of *Fxr1* leads to the accumulation of p21^Cip1^, and finally cellular senescence [92]. As an E3 ubiquitin ligase, Fbxo4 loss leads to the accumulation of Fxr1; while the ectopic expression of Fbxo4 reduces Fxr1 levels, resulting in the elevation of both p21^Cip1^ and p27^Kip1^ that can induce the senescence of HNSCC cells [1].

### 6.5. Other Biological Functions

Under glutamine-depleted conditions, the loss of Fbxo4 leads to the accumulation of cyclin D1, which at least partially enhances apoptosis in mouse embryonic fibroblasts, NIH3T3 cells and ESCC cells [51]. However, the overexpression of Fbxo4 destabilizes Mcl-1 in human NSCLC cells, resulting in apoptosis when exposing cells to chemotherapeutic compounds, such as Cisplatin or Paclitaxel [22]. These findings suggest that distinct effects of Fbxo4 on apoptosis are context-dependent, i.e., the effects depend on cell type or genetic background. As such, further studies are required to dissect the detailed molecular mechanisms. 

In addition, Fbxo4 also regulates the metastasis of breast cancer cells through controlling the stability of ICAM-1, which regulates the expression of epithelial–mesenchymal transition (EMT) markers, including E-cadherin (*CDH1*), Vimentin (*VIM*), Zinc finger E-box-binding homeobox 1 (*ZEB1*), and Snail Family Transcriptional Repressor 2 (*S**NAI2/S**LUG*) [25]. Overexpression of Fbxo4 not only reduces the size and weight of primary tumor xenografts, but also compromises the formation of metastatic tumors. Moreover, elevated Fbxo4 levels are correlated with longer survival in comparison to those with low Fbxo4 expression in breast cancer patients.

## 7. Conclusions

Generally, Fbxo4 functions as an E3 ubiquitin ligase that ubiquitylates and directs its substrates to the proteasome for degradation. All well-known Fbxo4 substrates can participate in a variety of biological processes related to tumor development and progression, defining Fbxo4 as a tumor suppressor. Due to the complexity of protein ubiquitylation, further studies are required to identify new substrates of Fbxo4 and whether Fbxo4 can catalyze the formation of other poly-ubiquitylation chains, such as Lysine 63-linked ones. This will help to draw a big picture of Fbxo4 functions under both physiological and pathological conditions that will specifically facilitate the start of a new precise medicine era through targeting Fbxo4 in human cancers.

## Figures and Tables

**Figure 1 cancers-14-02133-f001:**
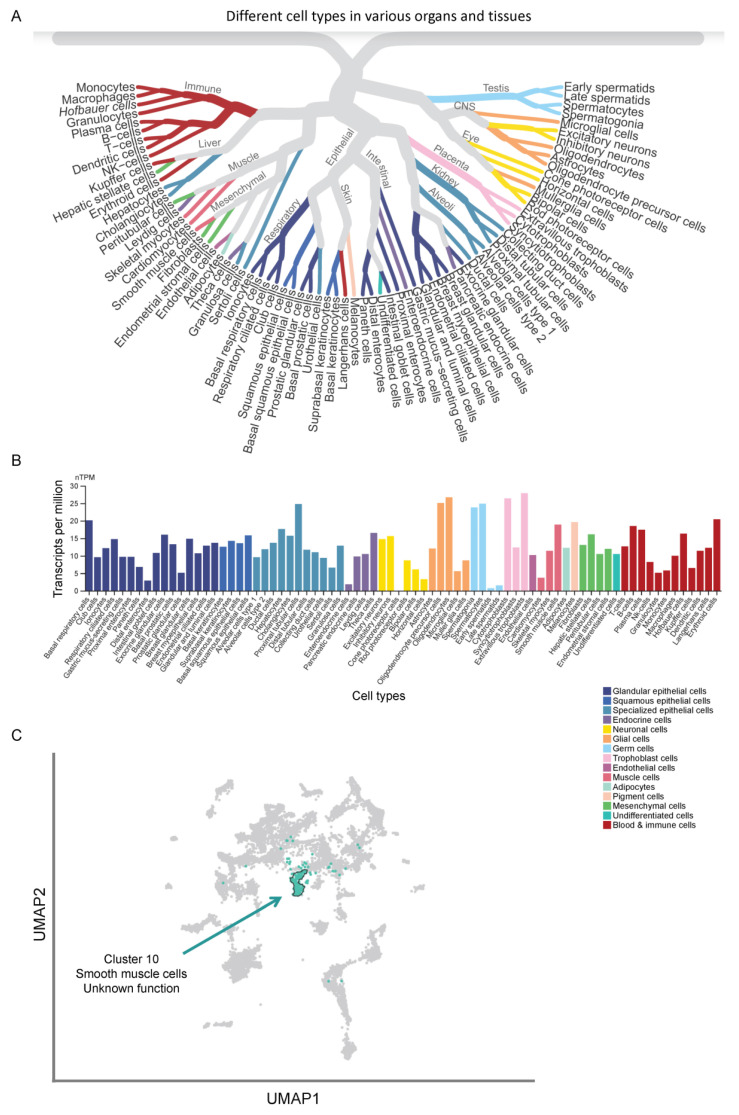
The single cell RNA sequencing identifies the expression pattern of *F**BXO4* in different cells (www.proteinatlas.org (accessed on 1 March 2022)). (**A**) The schematic illustration demonstrates the sequencing data are categorized into fifteen different cell type groups regarding to organ and tissue origins. (**B**) In general, there is low cell type specificity of *F**BXO4* expression among different cells according to that shown in Panel A. (**C**) The expression clustering analysis indicates *F**BXO4* belongs to cluster 10 “Smooth muscle cells-Unknown function” (highlighted by Cyan color). UAMP, Uniform Manifold Approximation and Projection.

**Figure 2 cancers-14-02133-f002:**
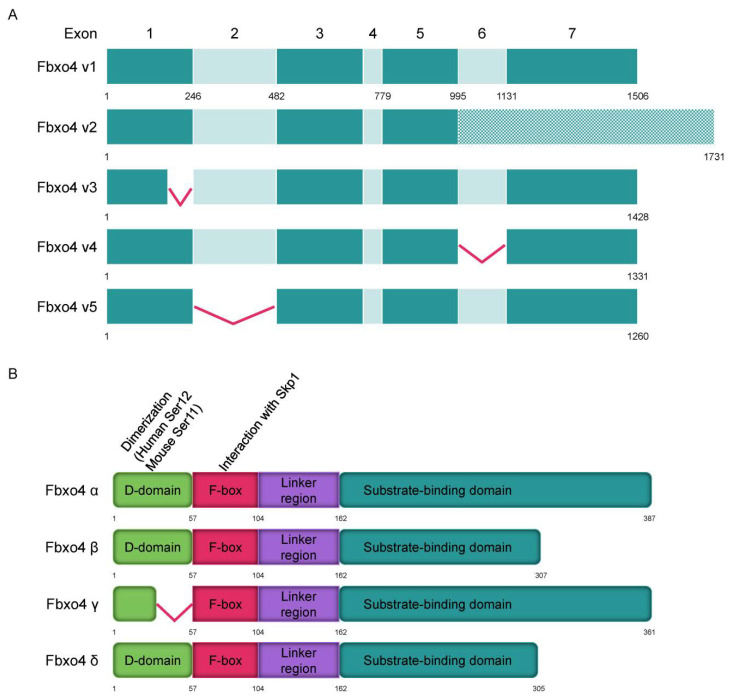
The transcript variants of *FBXO4* and protein structure of different isoforms. (**A**) Five transcript variants of *FBXO4* are shown in schematic plots. (**B**) Four different isoforms of Fbxo4 are identified with four domains: D domain for dimerization, F-box domain for forming SCF complex, Linker region, and substrate binding domain.

**Figure 3 cancers-14-02133-f003:**
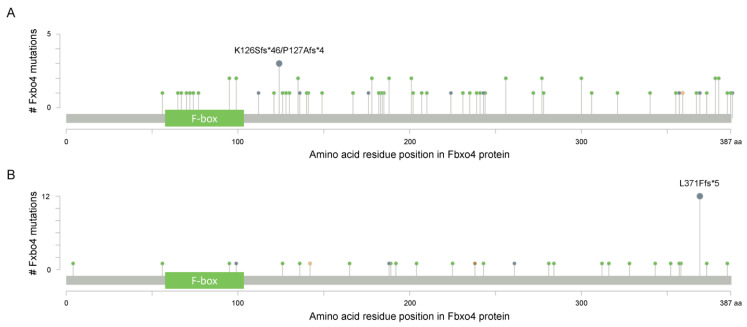
The cBioportal analysis reveals the existence of *FBXO4* mutations (www.cbioportal.org (accessed on 12 April 2022)). (**A**) The identified mutations of *FBXO4* in tumor tissues from the TCGA PanCancer Atlas Studies; all PanCancer related tumors are included for this analysis; there are 10,967 samples in total. (**B**) The identified mutations of *FBXO4* in cells from the Cancer Cell Line Encyclopedia (Broad, 2019); all cells in Cancer Cell Line Encyclopedia (Broad, 2019) are utilized for this analysis. *x*-axis indicates the amino acid residue position of Fbxo4 protein while y-axis indicates the number of mutations for each highlighted residues.

**Figure 4 cancers-14-02133-f004:**
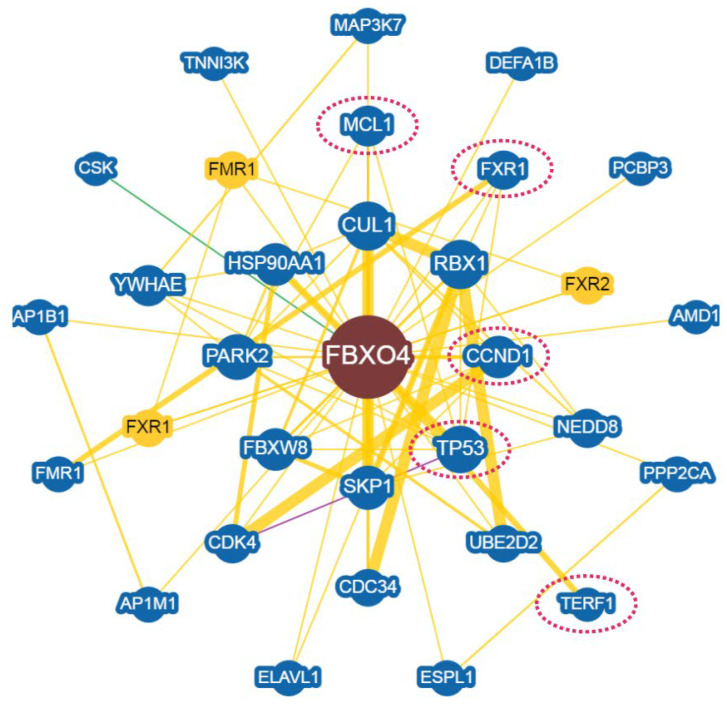
The BioGRID analysis suggests potential substrates of Fbxo4 (https://thebiogrid.org/ (accessed on 22 January 2022)). Yellow lines, Association with Physical Evidence; Green lines, Association with Genetic Evidence; Purple lines, Association with Genetic and Physical Evidence. Pink circle indicates the existence of biochemical data to support they are Fbxo4 substrates.

**Figure 5 cancers-14-02133-f005:**
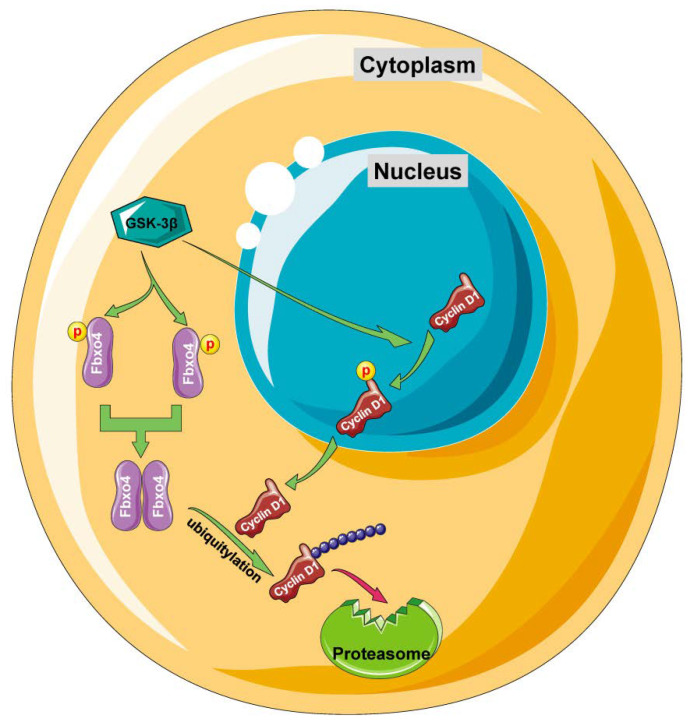
The detailed mechanism of how Fbxo4 mediates the ubiquitylation and degradation of cyclin D1. GSK-3β phosphorylates cyclin D1 at Thr-286 that facilitates its translocation from nucleus to cytoplasm. GSK-3β also phosphorylates Fbxo4 at Ser-12/Ser-11, promotes the homodimerization and activation of Fbxo4. In the cytoplasm, Fbxo4 catalyzes the ubiquitylation of cyclin D1 that targets it for proteasome-mediated degradation.

**Figure 6 cancers-14-02133-f006:**
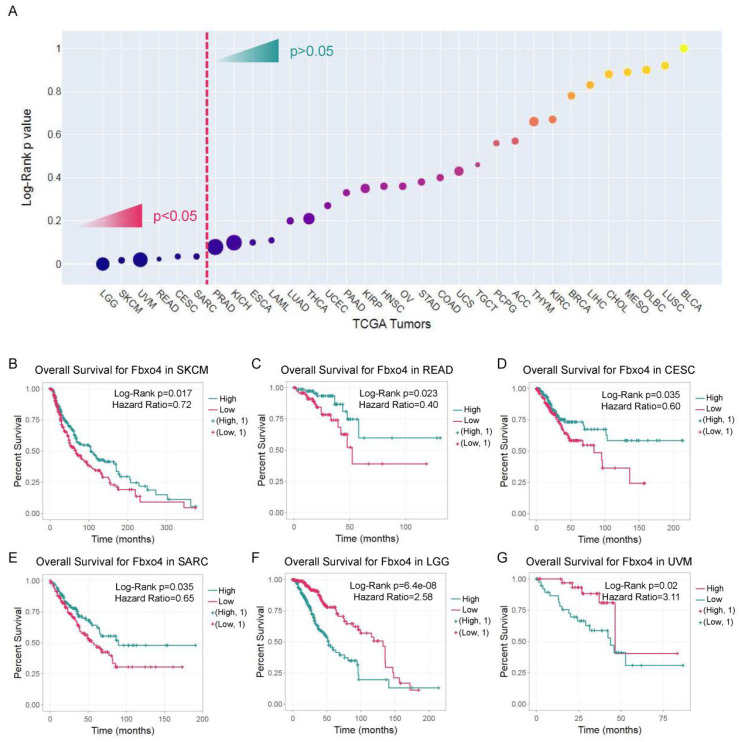
The prognostic importance of Fbxo4 in tumors from the human TCGA database. This analysis is processed by the ENCORI online tool (https://starbase.sysu.edu.cn (accessed on 12 April 2022)). (**A**) The summarization of Log-Rank *p* values of different TCGA tumors; y-axis indicates the *p* values and the circle size indicates the hazard ratio for each tumor. (**B**–**G**) Survival analysis based on Fbxo4 expression in patients with SKCM, READ, CESC, SARC, LGG and UVM. The cut-off of Log-Rank *p* value is setup at 0.05, and Hazard Ratio is shown in the figures.

**Table 1 cancers-14-02133-t001:** The clinicopathological importance of Fbxo4 in human cancers.

Cancers	SubStrates	Biological Functions of Fbxo4	Tumor Progression	References
ESCC	Cyclin D1	Fbxo4 compromises cell cycle progression, colony formation and oncogene-induced transformation	Various Fbxo4 mutations are identified in human ESCC samples	[21,27,44,108,112]
Melanoma	Cyclin D1	Fbxo4 I377M mutation causes the accumulation of cyclin D1 in melanoma cells	Loss of Fbxo4 increases tumor aggressiveness and reduces survival in Braf^V600E/+^ melanoma mouse model	[79]
HCC	Cyclin D1	Different Fbxo4 transcript variants perform different functions in regulating cyclin D1 expression	Fbxo4 is downregulated in human HCC tissues comparing to normal liver	[33]
Breast phyllodes tumors	Cyclin D1	Patients with Fbxo4 S8R mutation have elevated cyclin D1 levels	-	[109]
HNSCC	Fxr1	Fbxo4 suppresses cell proliferation and cellular senescence	Tumor tissues have low Fbxo4 levels relative to normal counterparts	[1]
Lung cancer	Mcl-1	Fbxo4 reduces cell survival	Reduced Fbxo4 expression leads to acquired resistance to chemotherapeutic compounds	[22]
Breast cancer	ICAM-1	Fbxo4 compromises EMT and tumor metastasis	Low Fbxo4 expression is associated with poor prognosis	[25,110]

## Data Availability

No new data were created or analyzed in this study. Data sharing is not applicable to this article.

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
