# Peer review of "The E3 Ubiquitin Ligase Fbxo4 Functions as a Tumor Suppressor: Its Biological Importance and Therapeutic Perspectives"

_cancers, 2022, doi:10.3390/cancers14092133_

Round 1

Reviewer 1 Report

In the manuscript submitted by Shuo Qie, current data on the E3 ubiquitin ligase Fbxo4 is reviewed. The review is well written and gives a good overview on the current understanding of Fbxo4 function in light of biological importance and therapeutic perspectives. However, the figures are poorly prepared and lack diligence.

Main point

  • The Figures should be better discussed within the text. In case of composite figures in the text it should be referred to Figure xa, Figure xb etc.
  • Figure legends are too brief to understand the Figure. No explanation about the axis is given. It does not become clear what is displayed.
  • Figure 1 is of poor quality and cannot be read
  • Figure 3 is of poor quality and poorly explained in the figure legend
  • Figure 4 What is the meaning of the yellow lines. Explain better in legend
  • Figure 6 is a complex figure with several parts. However, the text within the figures is not readable. What do we see in figure A? The figure legend is very brief. This figure is really poorly done

Minor points

Page 4

“It is common that FBPs have different isoforms …” What is the meaning of “common” in this context? Please rephrase.

Figure legend to figure 2: What is the meaning of “gene variants”. Are these different genes, or splice variants from one gene? Please carefully revise.

Author Response

In the manuscript submitted by Shuo Qie, current data on the E3 ubiquitin ligase Fbxo4 is reviewed. The review is well written and gives a good overview on the current understanding of Fbxo4 function in light of biological importance and therapeutic perspectives.

Reply: Thank you for your critical insight of my manuscript.

However, the figures are poorly prepared and lack diligence.

Reply: Thank you for pointing this out. Actually, I also find the same thing when I check the first version of my manuscript; particularly the figure resolution is really poor for the PDF version. In order to present the data better, I have made necessary changes of all the figures based on the reviewers’ suggestions.

Main point

  • The Figures should be better discussed within the text. In case of composite figures in the text it should be referred to Figure xa, Figure xb etc.

Reply: Thank you for your great suggestion. I have added more discussion in the main text to interpret the figure data; meanwhile, I also changed the presentation of different panels of figures wherever it is necessary.

  • Figure legends are too brief to understand the Figure. No explanation about the axis is given. It does not become clear what is displayed.

Reply: I totally agree with the reviewer for this point. I added the meaning of the axis wherever it is necessary.

  • Figure 1 is of poor quality and cannot be read

Reply: I have improved the quality of Figure 1.

  • Figure 3 is of poor quality and poorly explained in the figure legend

Reply: As being replied above, I have improved the quality of Figure 3 and added more introductions in the figure legend.

  • Figure 4 What is the meaning of the yellow lines. Explain better in legend

Reply: I have put more interpretation of the meaning of different lines in the figure legend. For example, Yellow lines, Association with Physical Evidence; Green lines, Association with Genetic Evidence; Purple lines, Association with Genetic and Physical Evidence.

  • Figure 6 is a complex figure with several parts. However, the text within the figures is not readable. What do we see in figure A? The figure legend is very brief. This figure is really poorly done

Reply: Thank you for your suggestion. I have changed this figure and put more information in the figure legend to make this figure is readable and support my conclusion.

Panel A indicates the Log-Rank p values of Fbxo4 in listed TCGA tumors from the PanCancer analysis on ENCORI website (https://starbase.sysu.edu.cn/panGeneCoExp.php#). P value cutoff is set up at 0.05 for statistical analysis, the circle size indicates the hazard ratio for each tumor.

I re-made other panels to increase the font size to make this figure readable. Another important thing is that I deleted several panels (Old Panels F, G, J & K) because the p value is more than 0.05. The reason I put them in the first version is to show the clinical importance of Fbxo4 in more tumors although the p values is more than 0.05. Since the font size is too small, I decided to delete these panels to leave more space to increase the font size.

Minor points

Page 4

“It is common that FBPs have different isoforms …” What is the meaning of “common” in this context? Please rephrase.

Reply: Thank you for pointing out this point. The “common” here means “frequently encountered”. That means “FBPs have different isoforms” is a usual phenomenon. I think it is a good idea to delete “It is common” as this may cause misunderstanding.

Figure legend to figure 2: What is the meaning of “gene variants”. Are these different genes, or splice variants from one gene? Please carefully revise.

Reply: Thank you for correcting this. This is my mistake. It indicates the different transcripts of Fbxo4 gene transcript. To address the reviewer’s concern, I have changed this figure legend.

Reviewer 2 Report

The paper provides overview of E3 ubiquitin ligase Fbxo4 molecular properties, recognized substrates, and potential function as a tumor suppressor. The relevant literature has been comprehensively reviewed. Author describes Fbxo4 functions with regard to its involvement in signaling pathways crucial for cancer development and progression. The paper topic is of potential clinical significance and elucidation of FBXO4 mutations role and/or dysregulated gene/protein expression may lead to the design of personalized anti-cancer therapies. However, several issues, mainly of a technical nature, should be clarified to improve the paper and make it more informative for the readers:

Major points:

  1. FBXO4 expression levels in certain types of cancers correlate with patient survival, however high expression of this “tumor suppressor to be” gene doesn’t necessarily correlate with better outcome. One should be aware that mediating shelterin (Trf1) and p53 degradation by Fbxo4 may increase genome instability; not to mention unknown effect of Fbxo4 on mutated p53, frequently observed in many cancer types. Moreover, different isoforms of Fbxo4 may have opposite effects on cancer cell division and migration etc., and in the consequence on cancer patient survival. Finally, information on gene expression levels available from TCGA/c-bioportal doesn’t take the occurence of a particular gene variants into account. The above issues could be more extensively discussed.
  2. The letters in paper’s Figures 1, 3 and 6 are impossible to read and it’s impossible to evaluate figures content. Please provide readable version of these Figures.
  3. It is unclear how Figure 1 has been created. The source data are not precisely cited and methods used to perform analysis are not described.

Minor points:

Single typos, like:

  1. Loss of function instead of loss of functions (page 5)
  2. Evidence instead of evident (page 6)
  3. Cytoplasm instead of cytoplasma (page 8, Figure 5)
  4. Mcl-1 subchapter number should be 3.4.5?

  1. Please provide the proper citation for databases used in the review: c-bioportal and BIOGRID.
  2. The chapter headers style is not consistent.
  3. Human gene names should be written in italics.
  4. The Patient Survival data not entirely fit the chapter they are located in – probably the separate chapter on the clinical data need to be added before the biological function (3.5).

Author Response

Reply to Reviewer #2

Comments and Suggestions for Authors

The paper provides overview of E3 ubiquitin ligase Fbxo4 molecular properties, recognized substrates, and potential function as a tumor suppressor. The relevant literature has been comprehensively reviewed. Author describes Fbxo4 functions with regard to its involvement in signaling pathways crucial for cancer development and progression. The paper topic is of potential clinical significance and elucidation of FBXO4 mutations role and/or dysregulated gene/protein expression may lead to the design of personalized anti-cancer therapies.

Reply: Thank you for your positive evaluation of my manuscript.

However, several issues, mainly of a technical nature, should be clarified to improve the paper and make it more informative for the readers:

Reply: I have made necessary changes to improve the presentation of my manuscript as suggested by the reviewer.

Major points:

  1. FBXO4 expression levels in certain types of cancers correlate with patient survival, however high expression of this “tumor suppressor to be” gene doesn’t necessarily correlate with better outcome. One should be aware that mediating shelterin (Trf1) and p53 degradation by Fbxo4 may increase genome instability; not to mention unknown effect of Fbxo4 on mutated p53, frequently observed in many cancer types. Moreover, different isoforms of Fbxo4 may have opposite effects on cancer cell division and migration etc., and in the consequence on cancer patient survival. Finally, information on gene expression levels available from TCGA/c-bioportal doesn’t take the occurrence of a particular gene variant into account. The above issues could be more extensively discussed.

Reply: I really appreciate the reviewer raised these concerns. Of course, I totally agree with the reviewer for this critical insight. To address these points and improve the quality of this manuscript, first, I put in another chapter to discuss the clinicopathological importance of Fbxo4 in human cancers; second, I present the TCGA data and emphasize the discrepancies of the role of Fbxo4 as a tumor suppressor; Third, I discuss more possibilities why the discrepancies exist in this part based on the reviewer’s suggestions as listed above.

  1. The letters in paper’s Figures 1, 3 and 6 are impossible to read and it’s impossible to evaluate figures content. Please provide readable version of these Figures.

Reply: Thank you for pointing out this. I have improved the quality of these figures. Several key points made: 1) increase the resolution, 2) enlarge the font size, and 3) add the title for each axis wherever necessary.  

  1. It is unclear how Figure 1 has been created. The source data are not precisely cited and methods used to perform analysis are not described.

Reply: This is my mistake. I forgot to add in the methods used and cite relevant references. Actually, this analysis was done using the Human Protein Atlas (www.proteinatlas.org). I also added the citation.

Minor points:

Single typos, like:

  1. Loss of function instead of loss of functions (page 5)

Reply: Thank you. I have changed this to “Loss of function”.

  1. Evidence instead of evident (page 6)

Reply: Thank you. I have changed this to “evidence”.

  1. Cytoplasm instead of cytoplasma (page 8, Figure 5)

Reply: Thank you. I have changed this to “cytoplasm”.

  1. Mcl-1 subchapter number should be 3.4.5?

Reply: Thank you. I have changed this.

  1. Please provide the proper citation for databases used in the review: c-bioportal and BIOGRID.

Reply: Thank you. I have added relevant citation for all the public databases (listed below).

ProteinAtlas [1,2]

cBioportal [3,4]

BioGrid [5]

ENCORI [6]

  1. Ponten, F.; Jirstrom, K.; Uhlen, M. The Human Protein Atlas--a tool for pathology. J Pathol 2008, 216, 387-393, doi:10.1002/path.2440.
  2. Karlsson, M.; Zhang, C.; Mear, L.; Zhong, W.; Digre, A.; Katona, B.; Sjostedt, E.; Butler, L.; Odeberg, J.; Dusart, P.; et al. A single-cell type transcriptomics map of human tissues. Sci Adv 2021, 7, doi:10.1126/sciadv.abh2169.
  3. Cerami, E.; Gao, J.; Dogrusoz, U.; Gross, B.E.; Sumer, S.O.; Aksoy, B.A.; Jacobsen, A.; Byrne, C.J.; Heuer, M.L.; Larsson, E.; et al. The cBio cancer genomics portal: an open platform for exploring multidimensional cancer genomics data. Cancer Discov 2012, 2, 401-404, doi:10.1158/2159-8290.CD-12-0095.
  4. Gao, J.; Aksoy, B.A.; Dogrusoz, U.; Dresdner, G.; Gross, B.; Sumer, S.O.; Sun, Y.; Jacobsen, A.; Sinha, R.; Larsson, E.; et al. Integrative analysis of complex cancer genomics and clinical profiles using the cBioPortal. Sci Signal 2013, 6, pl1, doi:10.1126/scisignal.2004088.
  5. Oughtred, R.; Rust, J.; Chang, C.; Breitkreutz, B.J.; Stark, C.; Willems, A.; Boucher, L.; Leung, G.; Kolas, N.; Zhang, F.; et al. The BioGRID database: A comprehensive biomedical resource of curated protein, genetic, and chemical interactions. Protein Sci 2021, 30, 187-200, doi:10.1002/pro.3978.
  6. Li, J.H.; Liu, S.; Zhou, H.; Qu, L.H.; Yang, J.H. starBase v2.0: decoding miRNA-ceRNA, miRNA-ncRNA and protein-RNA interaction networks from large-scale CLIP-Seq data. Nucleic Acids Res 2014, 42, D92-97, doi:10.1093/nar/gkt1248.

  1. The chapter headers style is not consistent.

Reply: Thank you for pointing this out. Obviously, those are some mistakes made during the editing process since the first version of my manuscript doesn’t contain these numeric headers. I have changed those headers to follow the logic flow of this review. Hopefully the following outline will help you to understand the revised version.

  1. Introduction
  2. The Fbxo4 gene and protein structure
  3. The regulation of Fbxo4 expression and activity

          3.1. Translational regulation of Fbxo4

          3.2. αB-crystallin as a co-factor of Fbxo4

          3.3. Phosphorylation and dimerization of Fbox4

  1. The identified substrates of Fbxo4

          4.1. Cyclin D1

          4.2. Trf1/ Pin2

          4.3. p53

          4.4. Fxr1

          4.5. Mcl-1

          4.6. ICAM-1

          4.7. PPARγ

  1. The clinicopathological importance of Fbxo4
  2. The biological functions of Fbxo4

          6.1. Cell cycle

          6.2. DNA damage response

          6.3. Tumor metabolism

          6.4. Cellular senescence

          6.5. Other biological functions

  1. Conclusions

  1. Human gene names should be written in italics.

Reply: I have italized the human gene names.

  1. The Patient Survival data not entirely fit the chapter they are located in – probably the separate chapter on the clinical data need to be added before the biological function (3.5).

Reply: Thank you for emphasizing this. I have added a separated topic and Table 1 to show the clinicopathological importance of Fbxo4 in tumors.

Reviewer 3 Report

Fbxo4, also known as Fbx4, is an F-box protein having a conserved F-box domain that belongs to the F-box protein family. Fbxo4 forms a complex with S-phase kinase-associated protein 1 and Cullin1 to accomplish its tasks, which needs the activation of RING-box protein 1 as a regulator. Fbxo4's biological functions suggest that it could be used to produce targeted therapeutics; hence, Fbxo4 consider a tumor suppressor. This review summarizes the Fbxo4 gene and protein structure, expression and activity, substrates, biological roles, and clinicopathological significance in human cancers. However, I have some reservations and recommendations.

It's also important to talk about the role of E3 ubiquitin ligase in anticancer resistance. I advise the authors to include this.

I recommend that the authors provide a table detailing the actions and mechanisms of E3 ubiquitin ligase Fbxo4 in various cancers.

The role of E3 ubiquitin ligase Fbxo4 in cancer immunotherapy should also be discussed.

English and grammar need to be improved.

Author Response

Reply to Reviewer #3

Comments and Suggestions for Authors

Fbxo4, also known as Fbx4, is an F-box protein having a conserved F-box domain that belongs to the F-box protein family. Fbxo4 forms a complex with S-phase kinase-associated protein 1 and Cullin1 to accomplish its tasks, which needs the activation of RING-box protein 1 as a regulator. Fbxo4's biological functions suggest that it could be used to produce targeted therapeutics; hence, Fbxo4 consider a tumor suppressor. This review summarizes the Fbxo4 gene and protein structure, expression and activity, substrates, biological roles, and clinicopathological significance in human cancers. However, I have some reservations and recommendations.

Reply: Thank you for commenting my review and helping with the improvement of this work. I have searched on PubMed and re-checked the literature based on the questions raised by the reviewer. For detailed information, please go to my point-by-point reply to each question.

It's also important to talk about the role of E3 ubiquitin ligase in anticancer resistance. I advise the authors to include this.

Reply: Thank you for pointing out this direction. By searching “Fbxo4 and resistance” on PubMed, not a lot of papers can be found. Therefore, I simply summarized and emphasized the role of Fbxo4 in anti-cancer resistance in the relevant paragraph discussing Fbxo4 substrates instead of putting in a new chapter.

One of my previous paper (Qie S, et al, Nat Commun, 2019) have revealed two major independent but intercorrelated findings: 1) dysregulation of Fbxo4-cyclin D1 drives glutamine addiction; 2) targeting glutamine-addiction can help overcoming acquired resistance to CDK4/6 inhibitor. According to the reviewer’s suggestions, I think it is a great direction for studying more on “Fbxo4 and anti-cancer resistance” in the future.

I recommend that the authors provide a table detailing the actions and mechanisms of E3 ubiquitin ligase Fbxo4 in various cancers.

Reply: Thank you for this great suggestion. I put Table 1 to introduce the clinicopathological importance of Fbxo4 in human cancers and mouse tumor models.

Table 1 The clinicopathological importance of Fbxo4 in human cancers

Cancers

Substrates

Biological functions of Fbxo4

Tumor progression

References

ESCC

Cyclin D1

Fbxo4 compromises cell cycle progression, colony formation and oncogene-induced transformation

Various Fbxo4 mutations are identified in human ESCC samples

[21,27,44,108,109]

Melanoma

Cyclin D1

Fbxo4 I377M mutation accumulates cyclin D1 in melanoma cells

Loss of Fbxo4 causes aggressiveness and reduced survival in BrafV600E/+ melanoma mouse model

[79]

HCC

Cyclin D1

Fbxo4 transcript variants perform different functions in regulating cyclin D1 expression

Fbxo4 is downregulated in human HCC tissues comparing to normal liver

[33]

Breast phyllodes tumors

Cyclin D1

Patients with Fbxo4 S8R mutation have elevated cyclin D1 levels

[110]

HNSCC

Fxr1

Suppresses cell proliferation and cellular senescence

Tumor tissues have low Fbxo4 levels relative to normal counterparts

[1]

Lung cancer

Mcl-1

Reduces cell survival

Reduced Fbxo4 expression leads to resistance to chemotherapeutic compounds

[22]

Breast cancer

ICAM-1

Compromises EMT and tumor metastasis

Low expression is associated with poor prognosis

[25,111]

The role of E3 ubiquitin ligase Fbxo4 in cancer immunotherapy should also be discussed.

Reply: To address the reviewer’s comment, I checked the PubMed by using “Fbxo4 and immune”, “Fbx4 and immune”, “Fbxo4 and immuno-” and “Fbx4 and immuno-”, unfortunately, I got no papers related to this topic. Therefore, it is impossible to add discussion on this topic. But this could be a very interesting topic for future investigation.

English and grammar need to be improved.

Reply: Thank you. I have sent the manuscript to a native speaker to help improving the writing. Please check the Track Changes in the main text.

Round 2

Reviewer 1 Report

Dear Shuo Qie,

I think that all critical point have been addressed in the revised manuscript. Quality has much improved.

Kind regards

Author Response

Reply to Reviewer #1 (2nd Round)

I think that all critical point have been addressed in the revised manuscript. Quality has much improved.

Reply: Thank you for your positive evaluation of my revised manuscript. Again, I made necessary changes to the English writing, for example, correcting the typo and grammar mistakes in the second revision.

Reviewer 2 Report

The implementation of reviewers suggestions significantly improved the manuscript quality. The graphs and diagrams are clear, adequately described, and easy to understand. Adding information on Fbxo4 involvement in different types of cancer in the form of a table, and discussing nuances of Fbxo4 function in cancer cells complete the picture.

In my opinion, only re-reading and some spell check is required, like in the Table 1: "Fbxo4 I377M mutation accumulates cyclin D1 in melanoma cells" - "leads to/causes accumulation of cyclin D1" instead of "accumulates"; "Loss of Fbxo4 causes aggressiveness... " - "increases tumour aggressiveness" instead of "causes aggressiveness..." etc.

After careful reading and small corrections, the manuscript is complete and ready for publication.

Author Response

Reply to Reviewer #2 (2nd Round)

The implementation of reviewers suggestions significantly improved the manuscript quality. The graphs and diagrams are clear, adequately described, and easy to understand. Adding information on Fbxo4 involvement in different types of cancer in the form of a table, and discussing nuances of Fbxo4 function in cancer cells complete the picture.

Reply: Thank you for your critical insight and positive comments on my revised manuscript.

In my opinion, only re-reading and some spell check is required, like in the Table 1: "Fbxo4 I377M mutation accumulates cyclin D1 in melanoma cells" - "leads to/causes accumulation of cyclin D1" instead of "accumulates"; "Loss of Fbxo4 causes aggressiveness... " - "increases tumour aggressiveness" instead of "causes aggressiveness..." etc.

Reply: I made changes to the above issues. Meanwhile, I also made necessary changes to other parts in order to correct the typo and grammar mistakes in the second revision.

After careful reading and small corrections, the manuscript is complete and ready for publication.

Reply: Thank you, again.

Reviewer 3 Report

All of the questions mentioned have been answered, and the current form of the article can be considered for acceptance.

Author Response

Reply to Reviewer #3 (2nd Round)

All of the questions mentioned have been answered, and the current form of the article can be considered for acceptance.

Reply: Thank you for your positive evaluation of my revised manuscript. Again, I made necessary changes to the English writing, for example, correcting the typo and grammar mistakes in the second revision.